# ViralRecall—A Flexible Command-Line Tool for the Detection of Giant Virus Signatures in ‘Omic Data

**DOI:** 10.3390/v13020150

**Published:** 2021-01-20

**Authors:** Frank O. Aylward, Mohammad Moniruzzaman

**Affiliations:** Department of Biological Sciences, Virginia Tech, Blacksburg, VA 24061, USA; monir@vt.edu

**Keywords:** giant viruses, nucleo-cytoplasmic large DNA viruses, metagenomics, endogenous viral elements, viral diversity

## Abstract

Giant viruses are widespread in the biosphere and play important roles in biogeochemical cycling and host genome evolution. Also known as nucleo-cytoplasmic large DNA viruses (NCLDVs), these eukaryotic viruses harbor the largest and most complex viral genomes known. Studies have shown that NCLDVs are frequently abundant in metagenomic datasets, and that sequences derived from these viruses can also be found endogenized in diverse eukaryotic genomes. The accurate detection of sequences derived from NCLDVs is therefore of great importance, but this task is challenging owing to both the high level of sequence divergence between NCLDV families and the extraordinarily high diversity of genes encoded in their genomes, including some encoding for metabolic or translation-related functions that are typically found only in cellular lineages. Here, we present ViralRecall, a bioinformatic tool for the identification of NCLDV signatures in ‘omic data. This tool leverages a library of giant virus orthologous groups (GVOGs) to identify sequences that bear signatures of NCLDVs. We demonstrate that this tool can effectively identify NCLDV sequences with high sensitivity and specificity. Moreover, we show that it can be useful both for removing contaminating sequences in metagenome-assembled viral genomes as well as the identification of eukaryotic genomic loci that derived from NCLDV. ViralRecall is written in Python 3.5 and is freely available on GitHub: https://github.com/faylward/viralrecall.

## 1. Introduction

Nucleo-cytoplasmic large DNA viruses (NCLDVs) are a group of dsDNA viruses of the phylum *Nucleocytoviricota* [1]. This group includes the largest viruses known, both in terms of physical dimensions and genome length [2]. Some members within the NCLDV group include those that infect metazoans and have been studied extensively, such as the *Poxviridae*, *Iridoviridae*, and *Asfarviridae*, while other families of NCLDVs, such as the *Phycodnaviridae*, *Mimiviridae*, *Marseilleviridae*, and *Pithoviridae*, have been discovered relatively recently and are known to infect protist lineages [3,4,5,6]. NCLDV genomes encode notably diverse functions that include many genes which are otherwise found only in cellular lineages, such as those involved in sphingolipid biosynthesis [7], amino acid metabolism [8], fermentation [9], glycolysis and the tricarboxylic acid (TCA) cycle [10], structuring of the eukaryotic cytoskeleton [11], and translation [12,13,14].

Numerous studies have begun to reveal that NCLDVs play key roles in the ecological and evolutionary dynamics of eukaryotes. Several cultivation-independent analyses have shown that NCLDVs are broadly distributed in the environment, and that they are particularly diverse and abundant in aquatic systems [10,15,16,17]. Moreover, analysis of eukaryotic genomes has revealed that endogenous NCLDVs are common in some lineages, and thereby play a significant role in host genome evolution [18,19,20,21,22]. There is, therefore, a need for bioinformatic approaches that facilitate these emerging frontiers of research in NCLDVs. Specifically, tools that can robustly identify signatures of endogenous NCLDVs in eukaryotic genomes will be useful for examining the role of NCLDVs in shaping eukaryotic genome evolution. Furthermore, multiple different approaches for binning NCLDV metagenome-assembled genomes (MAGs) from diverse environments have been developed [10,23,24], and approaches for quality-checking these results and removing possible contamination are needed given the unique features of NCLDV genomes.

The detection of NCLDV signatures in ‘omic data has proven challenging for a number of reasons. Firstly, NCLDVs are a diverse group of viruses that comprise several divergent lineages; the average amino acid identity between NCLDVs from different families can be as low as ~20%, and in some cases it can be difficult to identify any signatures of homology between disparate NCLDV genomes [10]. Tools such as MG-Digger [25], Giant Virus Finder [26], and FastViromeExplorer [27] can identify NCLDV sequences in metagenomic data using nucleotide-level homology searches, and these tools are useful for identifying NCLDVs that are closely related to reference genomes. Given the high degree of genomic diversity encompassed by NCLDV families, tools that leverage NCLDV-specific protein families are also needed to detect NCLDV sequences that are more divergent compared to those in reference databases. Secondly, NCLDV genomes often encode numerous genes of unknown function, or hypothetical genes for which it is difficult to infer evolutionary provenance based on functional annotations. Lastly, NCLDV genomes are chimeric in that they harbor genes with best matches to homologs present in bacteria, archaea, eukaryotes, and other viruses [28], and homology-based methods used to classify sequences from these genomes may therefore yield inconclusive results. Indeed, the presence of numerous genes in NCLDV genomes that have previously been identified only in cellular lineages complicates efforts to correctly classify NCLDV sequences.

In a previous study, we developed a workflow to identify viral regions in genomic data [18]. This workflow relied on searches against protein families present in the Viral Orthologous Groups database ([29]) to identify genomic loci that derive from viruses. The VOG database contains orthologous groups from numerous viruses; therefore, this tool could not discriminate between different viral groups, and several additional analyses were needed to trace the specific viral provenance of the sequences identified. In this study, we constructed a database of 28,696 giant virus orthologous groups (GVOGs) that represent protein families commonly found in NCLDVs. We also present ViralRecall v. 2.0, which leverages these NCLDV-specific protein families together with several additional features to permit the specific identification and annotation of NCDLV sequences in ‘omic data. We present several benchmarking criteria which establish that ViralRecall v. 2.0 has high specificity and sensitivity and can be used for several applications, including the removal of non-NCLDV contamination from metagenome-derived viral genomes and the identification of endogenous NCLDVs in eukaryotic genomes.

## 2. Materials and Methods

### 2.1. NCDLV Genomes Used for Database Construction

We compiled a database of 2908 NCLDV genomes, including metagenome-assembled genomes (MAGs) from two studies [10,23], and reference NCLDV genomes available in NCBI RefSeq as of June 1, 2020 [30]. We dereplicated these genomes by performing pairwise k-mer comparisons in MASH v. 2.0 (“mash dist” command with parameter -k 21, -s 300), and combining all genomes with a MASH distance of less than 0.05, which corresponded to ≥95% average nucleotide identity (ANI) [31]. NCLDV genomes were combined into clusters using single-linkage clustering, and the genome with the highest N50 contig size was used as the representative for that cluster. In this way we generated a nonredundant set of 2436 genomes that we subsequently used for downstream analyses (Appendix A).

### 2.2. Giant Virus Orthologous Groups (GVOGs)

To construct orthologous groups (OGs) we used a small subset of high-quality NCLDV genomes. We did this to eliminate possible contamination present in fragmented NCLDV MAGs and to reduce the computational load necessitated by large-scale OG construction. Genomes were only included if: 1) we found all four of the single-copy marker genes A32-atpase (A32), B-family DNA Polymerase (PolB), viral late transcription factor 3 (VLTF3), and superfamily II helicase (SFII); 2) we found no duplicate marker genes with < 90% amino acid similarity, which is indicative of the possible binning of multiple NCLDV genomes together; 3) the genome was composed of less than 30 contigs in total; and 4) the genome was not classified as “low quality” by the Schulz et al. study [23]. A total of 888 genomes met all of these criteria, and we calculated OGs from this set. We first predicted proteins using Prodigal v. 2.6.3 [32] (default parameters) and then generated OGs using Proteinortho v. 6.06 with the parameters “-e=1e-5 --identity=20 -p=blastp+ --selfblast --cov=50 -sim=0.95” [33]. Proteins belonging to OGs with >1 member were aligned using Clustal Omega (default parameters) and trimmed using Trimal (parameter -gt 0.05), and Hidden Markov Models (HMMs) were subsequently generated using the hmmbuild command in HMMER3 [34,35,36]. Using this approach, we generated 28,992 OGs, which we refer to as giant virus orthologous groups (GVOGs).

Annotations were assigned to GVOGs by comparing all proteins comprising each OG to the EggNOG 5.0 and Pfam. v. 32 databases using the hmmsearch command in HMMER3 (e-value cutoff of 1e-3 used for EggNOG, --cut_nc option used for Pfam) [37,38]. For each GVOG, an annotation was given only if >50% of the proteins in that GVOG had the same best match in the databases.

### 2.3. Calculation of ViralRecall Scores

The workflow implemented in ViralRecall is depicted in Figure 1. ViralRecall calculates scores for input sequences (heretofore referred to as contigs for simplicity) based on the number of encoded proteins that have matches in both the GVOG and Pfam v. 32 databases. Matches are determined using the hmmsearch command in HMMER3, and the e-value threshold is adjustable in ViralRecall. The scores for contigs are calculated based on the HMMER3 scores for each protein, and they are provided either as a mean for entire contigs, or as a rolling average that can be specified by the user.

We found that some GVOGs are commonly found in *Caudovirales* as well as NCLDVs, and we therefore normalized the HMMER score of these GVOGs to avoid false-positive detection. We did this for *Caudovirales* because: 1) through manual inspection of some NCLDV MAGs we had detected cases of likely *Caudovirales* contamination; and 2) jumbo bacteriophages belonging to the *Caudovirales* had the highest ViralRecall scores among non-NCLDV dsDNA viruses tested, suggesting that this group of viruses is the most likely source of false positive detections.

To identify GVOGs present in *Caudovirales*, we surveyed a set of 3012 *Caudovirales* genomes available in NCBI as of July 6, 2020; we predicted proteins from these genomes using Prodigal (default parameters) and identified GVOGs using hmmsearch with an e-value of 1e-5. For each GVOG, we calculated scores using the following equation:S_gvog_ = B (P_NCLDV_ + P_Caudo_/P_NCLDV_)(1)
where S_gvog_ is the final GVOG score, B is the score calculated by HMMER3, P_NCLDV_ is the proportion of NCLDV genomes with hits to that HMM, and P_Caudo_ is the proportion of *Caudovirales* genomes with hits to that model. In this way, GVOGs that are common in *Caudovirales* do not lead to high ViralRecall scores when they are detected.

Similarly, we normalize matches to Pfam domains present in either >= 1% of the 888 high-quality NCLDV genomes or the 3012 reference *Caudovirales* genomes using the equation:S_pfam_ = B (1−P_NCLDV_) + P_Caudo_/P_NCLDV_(2)
where S_pfam_ is the final Pfam score, B is the HMMER3 score, P_NCLDV_ is the proportion of NCLDV genomes with hits to that in HMM, and P_Caudo_ is the proportion of *Caudovirales* genomes with hits to that model. In this way, Pfam domains that are common in NCLDV do not lead to low ViralRecall scores when they are detected, and domains present in both *Caudovirales* and NCLDV are given a weight proportional to their relative occurrence in these viral groups. A ViralRecall score is then calculated for each ORF predicted by Prodigal using the equation:S_final_ = √S_gvog_ − √S_pfam_(3)

Scaling the scores by their square root is necessary to account for differences in overall HMM scores that are obtained for the GVOG and Pfam databases; because the Pfam database is made of domain-level HMMs, the scores obtained against this database are typically smaller, and outlier GVOG scores can therefore substantially skew the overall score without this normalization.

Given this scoring procedure, the presence of NCLDV-specific GVOGs will lead to higher scores, while the presence of Pfam domains that are typically not found in NCLDVs will lead to lower scores. Final scores for ViralRecall are provided either as a mean for each input contig or replicon, or as a rolling average across a window size that users can adjust (default = 15 ORFs). For simplicity, we use a value of 0 as the cutoff when discriminating between NCLDV and non-NCLDV sequences, but in principle other values could be used and may be appropriate depending on the situation and the balance between sensitivity and specificity that is desired.

### 2.4. Benchmarking

For benchmarking ViralRecall on non-NCLDV viral sequences, we used a database of 879 non-NCLDV dsDNA genomes downloaded from NCBI. These genomes were selected because they are reference dsDNA viruses with genomes listed on the ICTV Virus Metadata Resource ([39]) and their genomes were available on NCBI RefSeq. The GVOG and Pfam normalization scores had been generated by using reference *Caudovirales* genomes in NCBI; therefore, we did not use genomes from this group that were present in the Virus Metadata Resource. Instead, we used a set of 336 jumbo bacteriophages (*Caudovirales*) that have been reported [40]. Additionally, we did not include any *Lavidaviridae* (virophage) in this set, because these viruses parasitize giant viruses, and, in some cases, may exchange genes with them [41]. To generate pseudocontigs for benchmarking, we used the gt-shredder command in genometools ([42]).

To benchmark ViralRecall on NCLDV sequences, we used a set of 1548 genomes in the NCLDV database described above. This included all genomes except those used in the construction of GVOGs, because those would not provide an unbiased assessment of the sensitivity of ViralRecall in detecting NCLDV sequences. For benchmarking purposes, ViralRecall was run with default parameters, with the only exception that the -c flag was used to generate mean contig-level scores.

For illustrative purposes we selected the following five NCLDV genomes from diverse families and provide the results generated by ViralRecall (shown in Figure 3): Acanthamoeba castellanii Medusavirus [43], Emiliania huxleyi virus 86 [7], Pithovirus sibericum [44], M. separata entomopoxvirus [45], and Hyperionvirus [46]. We also selected the genomes of four non-NCLDV dsDNA viruses for this purpose; we chose the jumbo bacteriophages with the highest mean score, lowest mean score, and longest length of those tested (FFC_PHAGE_43_1208, M01_PHAGE_56_67, and LP_PHAGE_CIR-CU-CL_32_18, respectively), as well as the human herpesvirus 3 strain Dumas [47]. Lastly, we also show the profiles for Yaravirus [48], a virus of *A. castelanni* with unclear evolutionary provenance, and the Sputnik virophage [41]. In all cases, the viral genomes shown here were not used in the construction of the GVOG database or for score normalization, thus they provide an unbiased assessment of ViralRecall results. For manual inspection of proteins encoded in contigs derived from suspected contamination, we performed homology searches against RefSeq v. 93 using BLASTP+ [49].

To illustrate how ViralRecall can be used to identify NCLDV signatures in eukaryotic genomes, we analyzed the *Hydra vulgaris*, *Bigelowiella natans* and *Asterochloris glomerata* genomes. Previous studies have already established NCLDV signatures in these genomes [19,50,51], and our results therefore provide independent verification.

## 3. Results and Discussion

We analyzed 879 non-NCLDV genomes to assess the specificity of ViralRecall in detecting NCLDV sequences. Of these genomes, seven had mean scores >0, and could be considered false positives because they had net positive signatures of NCLDVs (Figure 2a). This provides an estimated specificity of 99.2% for ViralRecall when analyzing whole-genome data. Of the viruses with net positive scores, all had values <1 and were therefore only marginally positive. The seven genomes with positive scores included four *Alloherpesviridae*, one jumbo bacteriophage, one *Portogloboviridae*, and one *Lipothrixviridae* (Appendix A). A variety of other dsDNA viral groups, including *Adenoviridae*, *Baculoviridae*, and *Herpesviridae*, always had scores <0. Metagenome-derived viral genomes may often be fragmented; therefore, we also evaluated the false positive rate of ViralRecall when analyzing genome fragments. For this, we ran this tool on a set of 2973 non-overlapping pseudocontigs 5–100 kbp in length that we generated from the non-NCLDV dsDNA dataset. Of these pseudocontigs, 142 had positive scores, leading to a false-positive rate of 4.8% (specificity of 95.2%) (Appendix A). Of the pseudocontigs with positive scores, 66 (46%) were less than 20 kbp in length, and 114 (79.7%) belonged to jumbo bacteriophage, suggesting that fragmented sequences of *Caudovirales* sometimes have misleadingly high NCLDV signatures.

Assessing the sensitivity, or false-negative rate, of ViralRecall is complicated by the fact that 888 high-quality NCLDV genomes were used to construct the GVOGs used by this tool, and are therefore not appropriate to use for benchmarking purposes. Of the remaining 1548 NCLDV genomes that we compiled, most were derived from metagenomes, and possible contamination by non-NCLDV sequences could not be ruled out. Nonetheless, analysis of the ViralRecall scores of these 1548 NCLDV does provide some insight into the sensitivity of this tool for detecting true NCLDV sequences, and we provide the results for the 38,886 contigs of these genomes in Figure 2b. Of the 38,896 contigs, 657 (1.7%) had negative values (median of 9.7) indicating that ViralRecall is generally accurate in identifying NCLDV signatures in these sequences. Moreover, the 657 sequences with negative values comprise only 0.78% of the total bp in the dataset. Of the 657 sequences that had negative values, some likely derive from contamination, but others may represent real false negatives. It is notable that all sequences from cultivated viruses in this plot had positive values; all of the negative values derived from metagenome-derived genomes where contamination is more likely. Regardless, even if we assume that contigs with negative values are false negatives, we still arrive at a sensitivity of 98.3%.

We also examined the rolling average of ViralRecall scores on a subset of NCLDV and non-NCLDV genomes tested (Figure 3). NCLDV from diverse families generally show consistently high scores throughout the chromosomes. This includes Acanthamoeba polyphaga Medusavirus, which has been proposed to belong to a novel NCLDV family, as well as Pithovirus sibericum (Pithoviridae), Entomopoxvirus (Poxviridae), and Hyperionvirus (Mimiviridae). The rolling average for Emiliania huxleyi virus 86 was generally high, although it dipped below zero in two regions >10 kbp in length. We also note that ViralRecall recovered an overall positive for Yaravirus, a eukaryotic virus that may be a divergent member of the NCLDV family [48]; ViralRecall identified 14 GVOG hits out of 70 predicted proteins, providing further evidence that even divergent NCLDVs or related groups can be detected with this tool. Lastly, we recovered a consistently high score for the sputnik virophage (family Lavidaviridae); this is not unexpected given that this virus parasitizes Mimivirus and has likely exchanged genes with NCLDV [41].

Overall, these results demonstrate that ViralRecall has high specificity and sensitivity at recovering NCLDV sequences, but this tool should still be used alongside other independent methods for sequence classification. This is because sequences derived from NCLDV, especially if they are fragmented into short contigs, may not contain enough predicted proteins with matches in the GVOG database to produce a positive overall score. Similarly, other viral groups such as large *Caudovirales* (jumbo phage) can have a misleadingly high number of GVOG matches. For these reasons, it would still be advisable to use alternative approaches when determining the provenance of sequences in ‘omic datasets, such as phylogenetic analysis of marker genes and homology searches against reference databases. To assist with manual inspection of results, ViralRecall also searches against 10 custom HMMs for NCLDV marker genes and reports the presence of hits to these protein families in the output files. These genes include the five markers commonly used in phylogenetic reconstruction (PolB, A32, VLTF3, SFII, and MCP) as well as five others that are known to be present in many NCLDV genomes (RNA polymerase subunits (RNAPL and RNAPS), the mRNA capping enzyme (mRNAc), ribonucleotide reductase (RNR), and D5 primase/helicase (D5)) [10]. Inspection of the occurrence of these markers can be useful for determining if a sequence derives from NCLDV; for example, of the seven non-NCLDVs that had scores >0, none encoded proteins with matches to MCP, A32, VLTF3, mRNAc, PolB and D5 markers, although divergent hits with low scores were observed for SFII, and RNR homologs. None of these seven non-NCLDVs had hits to the RNA polymerase subunits RNAPL or RNAPS, but identification of these markers without subsequent phylogenetic analysis cannot be considered evidence of NCLDV provenance given the universal presence of these protein families in cellular life as well as some *Caudovirales* [52].

The overall incidence of potential contamination in the 1548 NCLDV genomes was low (657 out of 38,896 contigs with scores <0). The majority of the NCLDV genomes in our database were constructed from metagenomes, which suggests that current approaches that have been employed for binning NCLDVs have successfully removed most non-NCLDV contaminations during the binning process. Nonetheless, we did identify some cases of contamination that highlight how ViralRecall may be useful for quality-checking MAGs. For this, we manually examined two NCLDV MAGs that contained the longest contigs with scores <0. We provide plots of the ViralRecall rolling averages for these MAGs in Figure 4a, and a dot plot of the mean contig scores in Figure 4b. A 123 kbp contig in GVMAG-S-1064190.84 together with a shorter 6 kbp contig had a scores <0, and we performed subsequent homology searches against RefSeq v. 93 which confirmed that the large contig bears signatures of *Caudovirales* origin, including the presence of phage wedge, baseplate, head and tail proteins. The ERX556094.26 MAG contained a large 68 kbp contig with negative ViralRecall score in addition to eight smaller contigs ranging from 13.3–27 kbp in length that also had negative scores. We performed homology searches of encoded proteins in these contigs against RefSeq, which revealed numerous hits to marine *Flavobacteriaceae*. The metagenome ERX556094 was generated from seawater, so the presence of marine *Flavobacteriaceae* in this sample was not unexpected. Analysis of these two NCLDV MAGs provided evidence that ViralRecall can be useful in identifying and removing contamination from non-NCLDV sources from NCLDV MAGs.

To assess the efficacy of ViralRecall in detecting signatures of NCLDV in eukaryotic genomes, we analyzed the genomes of *Hydra vulgaris*, *Bigelowiella natans,* and *Asterochloris glomerata* (Figure 5), which have previously been shown to harbor signatures of endogenous NCLDVs [19,50,51]. In *H. vulgaris*, we identified one NCLDV region that encompassed the entire 396 kbp contig NW_004166914.1 (Figure 5), consistent with previous findings. This region also encoded proteins with matches to the D5, MCP, RNAPL, RNAPS, and VLTF3 markers. For *B. natans*, we found 35 putative viral regions; the longest was a 230 kbp region in contig ADNK01000550.1 that also encoded proteins with matches to the MCP and A32 markers (Figure 5). At the beginning of this contig, we also found a 22.6 kbp region with NCLDV signatures. Lastly, for *A. glomerata*, we found 27 putative viral regions; the largest was a 224 kbp region in scaffold 80 (full length 1.2 of megabases) that encoded homologs to the MCP, A32, SFII, PolB, and D5 marker genes. These results highlight the variable architecture of endogenous NCLDVs; although some encompass entire contigs in draft genome assemblies, some span only smaller regions of chromosomes.

Overall, our results provide evidence that ViralRecall can effectively identify signatures of NCLDV in a variety of ‘omic datasets, but some limitations remain. For example, non-NCLDV sequences belonging to virophages and sometimes *Caudovirales* can have misleadingly high ViralRecall scores, and additional analyses will often be necessary to robustly classify these sequences. Other approaches, such as those that leverage ribosomal binding site motifs, have successfully been used for the identification of NCLDV sequences [23], and these methods could be used in addition to ViralRecall to confirm results. Lastly, ViralRecall relies on large-scale HMM searches against the Pfam and GVOG databases, which are time-consuming and may not be feasible for large datasets. For example, the application of ViralRecall to assembled metagenomic datasets may not be advisable; rather, the binning of putative NCLDV MAGs and subsequent analysis of those bins only is likely to be more efficient. Nonetheless, the benchmarking results we present are promising, and we anticipate that this tool will be useful for exploring the signatures of NCLDV in diverse ‘omic datasets and thereby facilitate studies that expand our knowledge of the ecological and evolutionary roles of these viruses in the biosphere.

## Figures and Tables

**Figure 1 viruses-13-00150-f001:**
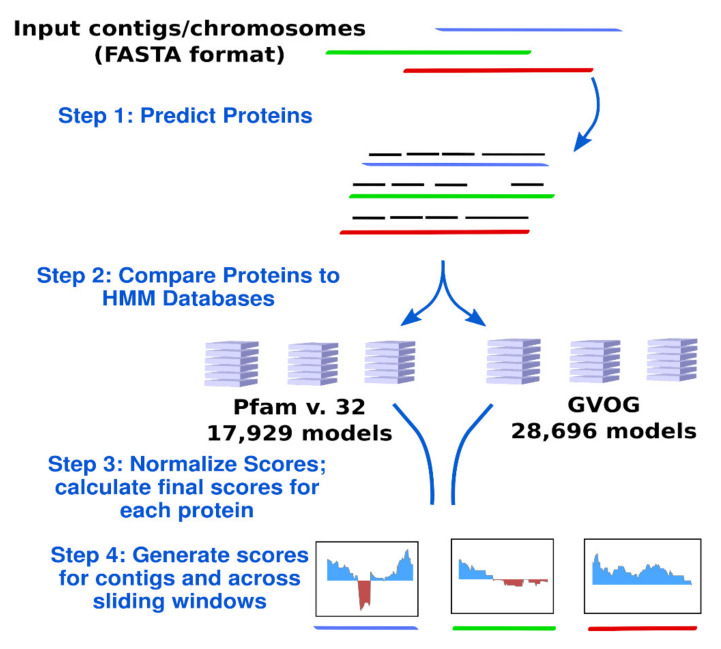
Diagram of the ViralRecall workflow. Abbreviations: GVOGs, giant virus orthologous groups.

**Figure 2 viruses-13-00150-f002:**
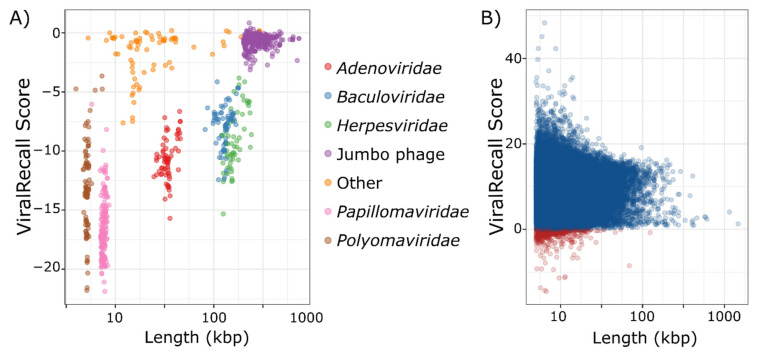
(**A**) ViralRecall scores and lengths for 879 non-nucleocytoplasmic large DNA viruses (NCLDV) dsDNA viruses. (**B**) ViralRecall scores and lengths for 38,886 giant virus contigs from 1548 reference and metagenome-assembled giant virus genomes. Contigs with scores <0 are colored red, while those with scores ≥0 are colored blue.

**Figure 3 viruses-13-00150-f003:**
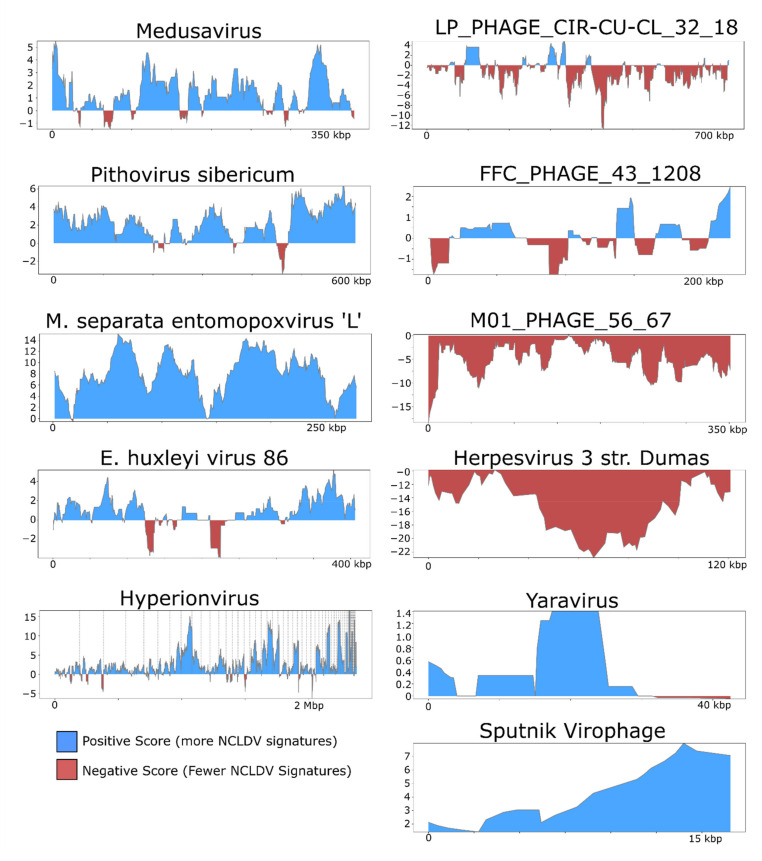
ViralRecall plots of diverse dsDNA viruses. NCLDV genomes are shown on the left, while the right panels show other dsDNA viruses, or highly divergent NCLDV in the case of Yaravirus. The jumbo bacteriophages LP_PHAGE_COMPLETE_CIR-CU-CL_32_18, FFC_PHAGE_43_1208, and M01_PHAGE_56_67 were chosen because they have the longest length, highest score, and lowest score, respectively, among the 336 jumbo phages tested.

**Figure 4 viruses-13-00150-f004:**
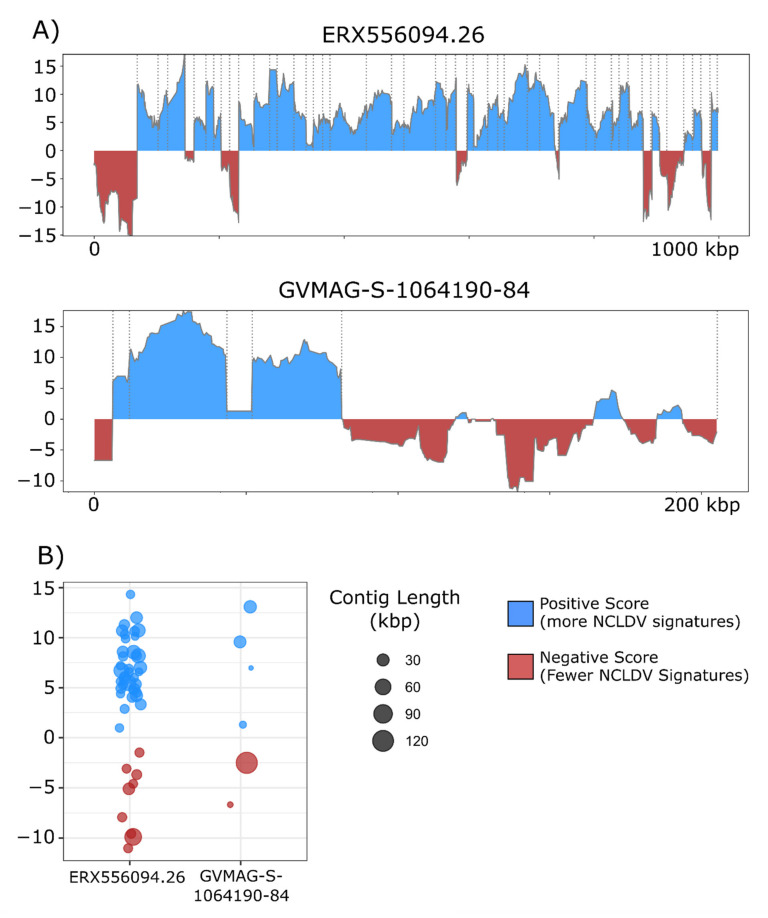
(**A**) ViralRecall plots for the giant virus MAGs ERX556094.26 and GVMAG-S-1064190.84 demonstrating that both contain non-NCLDV contamination. For ERX556096-26, nine contaminant contigs were detected, while two were found in GVMAG-S-1064190.84. (**B**) Dot plot of the mean ViralRecall scores for all contigs in ERX556094.26 and GVMAG-S-1064190.84. Contigs with ViralRecall scores < 0 are colored red, and dot sizes are proportional to contig size.

**Figure 5 viruses-13-00150-f005:**
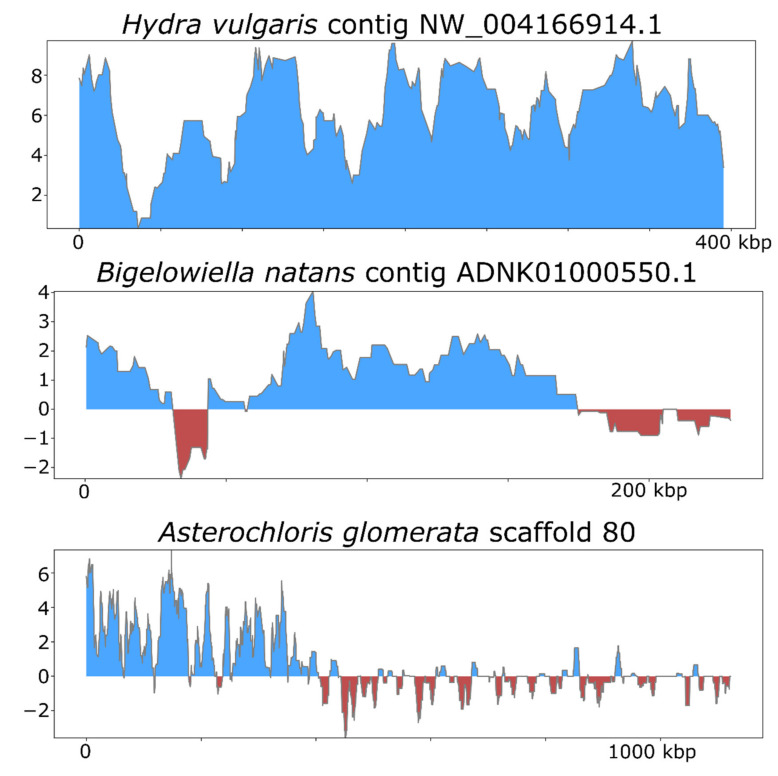
ViralRecall plots for endogenized viral regions identified in *Hydra vulgaris*, *Bigelowiella natans*, and *Asterochloris glomerata*.

## Data Availability

All codes are available at https://github.com/faylward/viralrecall.

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
