# Peer review of "ViralRecall—A Flexible Command-Line Tool for the Detection of Giant Virus Signatures in ‘Omic Data"

_viruses, 2021, doi:10.3390/v13020150_

Round 1

Reviewer 1 Report

This study present a command based tool, VrialRecall, for identifying NCLDV genes with high sensitivity and specificity. The results for applying this tool were well demonstrated with both NCLDV, and non-NCLDV genomes. In addition to positive outcomes, the authors had also stated the limitations on this tool. Lastly the VrialRecall tool has been made available online and tutorials on its installation and usage are provided. Overall, this is a well-written manuscript and this tool sounds useful for the researchers. Therefore, this manuscript should be consider to be published in Viruses. 

Author Response

Thank you for your kind review.

Reviewer 2 Report

This manuscript presents ViralRecall - a bioinformatics tool for the detection of giant virus signatures.  Giant viruses belong to a group named Nucleo-Cytoplasmic Large DNA Viruses (NCLDV).  NCLDV is a diverse group of dsDNA viruses with the largest and most complex viral genomes that infects protists, animals and single celled algae. The authors accomplished a great deal in assessing specificity and sensitivity of ViralRecall at recovering NCLDV sequences and demonstrating that ViralRecall can effectively identify NCLDV sequences in eukaryotic genomes.

The manuscript can be published in its present form after some minor corrections:

The authors cite in the text Figure 1b, but there is no fig. 1b presented (I assume that the authors meant fig. 2b).

The Figure 2b isn’t mentioned in the main text.

Figure 5a is cited in the main text but figure 5a is absent.

Author Response

Thank you for your kind review and for spotting the typos regarding the figure labels. These have been corrected. The reference to Figure 1b should have been 2b. Figure 5 does not have panels, so we have removed the reference to "5a" and changed it to "5".

Reviewer 3 Report

In this article, the authors develop a command-line tool known as VirusRecall for the detection of giant viral signatures in the metagenomics data. The two objectives of the tool are the identification of NCLDV signatures in the environment sample and endogenous NCLDVs in eukaryotic genomes.

With a large number of giant viruses discovered in the recent past and many more are waiting to be discovered, this tool will be certainly helpful for the routine screening of the metagenomics data for the detection of giant-viral/NCLDV signatures. This is a well-written manuscript and shortcomings of the tool have been clearly articulated.

Some minor comments that need to be addressed. 

The two open-source pipelines/work-flows that are available currently, namely, MG-Digger and Giant Virus Finder, are useful for automated analysis of metagenomes for the presence of giant virus signatures.  Surprisingly these tools have not been discussed or referred to. I guess the VirusRecall tool has an added advantage of detecting the endogenous NCLDVs in eukaryotic genomes. Authors can briefly discuss in the introduction (or results and discussion) how their tool is different from these two.

Line 189- I think it should be Figure 2b instead of Figure 1b

Author Response

Thank you very much for mentioning MG-Digger and Giant Virus Finder. These tools are a bit different because they rely on nucleotide-level homology searches and typically use raw reads as input. Also we originally felt ViralRecall was distinct since it is generally applied to assembled metagenome bins or endogenous viruses, and therefore has a slightly different goal. However, we can see that citing and discussing these two other tools is appropriate, and we have included the following sentence in the introduction:

"Tools such as MG-Digger [26] and Giant Virus Finder [27] have been developed to identify NCLDV sequences in metagenomic data using nucleotide-level homology searches, and these tools are useful for identifying NCLDV that are closely related to reference genomes. Given the high degree of genomic diversity encompassed by NCLDV families, tools that leverage NCLDV-specific protein families are also needed to detect NCLDV sequences that are more divergent compared to those in reference databases."

Also, thank you for pointing out the typo regarding Figure 1b. This has been corrected.